# Discrepancy of Effective Water Diffusivities Determined from Dynamic Vapor Sorption Measurements with Different Relative Humidity Step Sizes: Observations from Cereal Materials

**DOI:** 10.3390/foods12071470

**Published:** 2023-03-30

**Authors:** Xuewei Zhao, Xiaoxiao Wei, Hongwei Wang, Xingli Liu, Yanyan Zhang, Hua Zhang

**Affiliations:** 1College of Food and Bioengineering, Zhengzhou University of Light Industry, Zhengzhou 450001, China; zhao_xuewei@zzuli.edu.cn (X.Z.); wxx@email.zzuili.edu.cn (X.W.); hwwang@zzuli.edu.cn (H.W.); 2017085@zzuli.edu.cn (X.L.); 2016038@zzuli.edu.cn (Y.Z.); 2Henan Key Laboratory of Cold Chain Food Quality and Safety Control, Zhengzhou 450001, China; 3Food Laboratory of Zhongyuan, Zhengzhou University of Light Industry, Zhengzhou 450001, China

**Keywords:** cereal, water sorption, drying, diffusivity, mass transfer, kinetics

## Abstract

Water diffusivity, a critical parameter for cereal processing design and quality optimization, is usually concentration-dependent. dynamic vapor sorption (DVS) system provides an approach to establishing the relationship between water concentration and diffusivity. However, the usual relative humidity (RH) jump during practical sorption processes is usually greater than that adopted in DVS measurements. Water vapor sorption kinetics of glutinous rice grains, glutinous rice flour and wheat flour dough films were measured using the DVS system to verify if varying RH step sizes can obtain identical diffusivities within the same range. The effective diffusivities were determined according to Fick’s second law. The results revealed that increasing RH step size led to a higher estimated diffusivity, regardless of whether the water concentration gradient or potential chemical gradient was considered a driving force for water diffusion. This finding was further confirmed by a linear RH scanning DVS measurement. The water concentration-dependent diffusivity obtained from a multi-step DVS measurement, according to Fick’s second law, will overestimate the required time for practical cereal drying or adsorption. Thus, this paradoxical discrepancy needs a new mass transfer mechanism to be explained.

## 1. Introduction

Cereals and cereal-based food products can either gain or lose water when exposed to a humid or dry environment. Water diffusion is a crucial step in controlling the adsorption or desorption process. Hence, the diffusion coefficient is necessary for modeling and designing various cereal processing operations such as drying, rehydration and storage [1]. This coefficient is usually obtained from kinetic data of three processes: drying, sorption and permeation [2,3]. The dynamic vapor sorption (DVS) method is an effective technique used to determine water diffusivities of cereal grains [4] and cereal-based foods [5,6,7] due to its convenient and precise control of both relative humidity (RH) and temperature.

Fick’s second law is commonly used to model water sorption kinetics and extract water diffusion coefficients [7,8]. Other than an instant equilibrium boundary condition, external mass transfer resistance is considered in some reports in order to improve modeling fitness [5,6]. An analytical solution to Fick’s second law can be obtained when certain assumptions are made, such as a constant diffusivity and the Dirichlet boundary conditions, for a well-defined geometry such as a cylinder, plate or sphere [9]. Otherwise, a numerical method is necessary for solving the conservation equation, and inverse optimization may be required to obtain the water diffusivity [10].

When water concentration does not change too much during a sorption step, water diffusivity can be considered constant to facilitate deriving water diffusivity from water sorption kinetic data. Otherwise, the diffusivity should be considered as a function of water concentration. Pre-assumed water concentration–diffusivity relations are sometimes assigned [11,12], but alternative methods have also been proposed, such as the regular regime approach [13] and Yamamoto’s model [14], to determine the concentration-dependent effective diffusivity from their desorption curves. An observer-based methodology recently presented by Vera et al. [15] offers another method for the same purpose. Jian and Jayas [16] treated diffusivity as a transient parameter and determined its time-dependent values from numerical simulation. Usually, these methods are tedious in operations.

The multi-step DVS test provides a convenient approach to evaluate the variation of diffusivity with water concentration. This approach usually selects an RH step size of 10% during DVS measurements [5,6,7], with diffusivity considered constant in each RH step and extracted according to Fick’s second law. After a multi-step continuous DVS test, a water concentration–diffusivity relation can be established. This relation is often displayed in a chart [6,7,11,17] or further mathematically treated to obtain an explicit diffusivity function of water content [18].

The RH jump is usually much greater than 10% during practical sorption processes when cereal is exposed to storage or drying [1,13,14,16]. Then, it is not clear whether the water concentration–diffusivity relation obtained from a multi-step DVS measurement is suitable for calculating the actual water adsorption or drying process. In other words, when different RH step sizes are imposed during sorption kinetic measurements, can equal water diffusivity be obtained at the same RH level?

In this work, sorption kinetics of the glutinous rice grains, glutinous rice flour and wheat dough films were measured under different RH step sizes using the DVS method, and water diffusivities were reversely estimated according to Fick’s second law. The objective of this study is to test the consistency of water diffusivities from DVS measurements with varying RH step sizes. These observations were further validated when the potential chemical gradient was considered a diffusion driving force, and a linear RH scanning procedure was used for DVS measurements.

## 2. Diffusion Models

Equation (1) is used to describe the mass conservation of water, with *c* representing water concentration (kg/m^3^), *t* for time (s), *J* for water flux (kg/(m^2^/s)) and ∇ for the Hamilton operator. Assuming molecular diffusion as the sole means of mass transfer and water concentration gradient as the only driving force, the *J* can be calculated according to Fick’s first law (Equation (2)), where *D* is the diffusion coefficient (m^2^/s).
(1)$$\frac{\partial c}{\partial t}+\nabla \cdot (J)=0$$
(2)J=−D∇c

Initial and boundary conditions are necessary for solving a partial differential equation. The initial water concentration can be assumed to be homogeneous in space and equal to the initial water concentration (*c*_0_) of each RH step during DVS measurements:(3)$$c=c_{0}\;\;\mathrm{at}\;t\;=\;0$$

There is no mass flux through the center of a sphere and the symmetry axes of a cylinder and a plate, and thus the boundary conditions at these locations are defined by Equation (4). Although an instant equilibrium boundary is commonly used in the literature [17,18,19], convective mass transfer resistance was considered in the present work. The boundary conditions at the exposed surface are defined by Equation (5).
(4)−n⋅(−D∇c)=0
(5)$$-n\cdot \left(-D\nabla c\right)=h_{m}\left(c_{v,\mathrm{surf}}-c_{v,\mathrm{air}}\right)$$
where *h*_m_ is the convective mass transfer coefficient (m/s). For DVS tests, its measured value was 8 × 10^−3^ m/s [20], and it was adopted in the present work. On the other hand, *c*_v.air_ is the vapor concentration of airflow in the DVS chamber (kg/m^3^), *c*_v.surf_ is the vapor concentration in the air equilibrated with the water concentration at the surface of a specimen (kg/m^3^). Assuming air is an ideal gas, the state equation $P_{v}V=nRT$ holds, where *P*_v_ is the partial pressure of vapor (Pa), *V* is the gas volume (m^3^), *n* is the mole number of vapor molecules in the *V* volume of gas, *R* is the general gas constant (8.314 J/(mol·K)), *T* is the temperature (K). Then, $c_{v}=\frac{nM}{V}=\frac{MP_{v}}{RT}=\frac{Ma_{w}P_{v,\mathrm{sat}}}{RT}$; here, *M* is water molecular weight (18 g/mol), *a*_w_ is water activity, = *RH*/100, *P*_v,sat_ is saturated pressure of vapor (Pa). By applying this information, Equation (5) can be transformed into Equation (6).
(6)$$-n\cdot (-D\nabla c)=\frac{h_{m}MP_{v,\mathrm{sat}}}{RT}\left(a_{w,\mathrm{surf}}-\frac{RH_{\mathrm{air}}}{100}\right)$$
where *RH*_air_ is the relative humidity of air (%), and *a*_w,surf_ is water activity equilibrated with *c*_v.surf_ according to the isotherm model.

### 2.1. For a Finite Cylinder

Water diffusion in a finite cylinder can be simplified to a two-dimension problem if water molecule movement occurs only in the axial and radial directions. In this case, ∇=∂/∂r+∂/∂z, where *r* and *z* are radial and axial distances, respectively. Equation (7) is obtained from Equation (1) combined with Equation (2) after some mathematical treatments.
(7)$$\frac{\partial c}{\partial t}=\frac{1}{r}\frac{\partial }{\partial r}(rD\frac{\partial c}{\partial r})+\frac{\partial }{\partial z}(D\frac{\partial c}{\partial z})$$

### 2.2. For a Sphere

In this work, the rice flour particles were assumed to be spherical and represented by a sphere with an average diameter of the particles. In order to reduce the workloads of computation, a simplified geometry of a line can be used if the water concentration is assumed dependent only on the radial distance *r*. In this case, ∇=∂/∂r, and the diffusion conservation equation (Equation (1)) takes the form:(8)$$\frac{\partial c}{\partial t}=\frac{1}{r^{2}}\frac{\partial }{\partial r}\left(r^{2}D\frac{\partial c}{\partial r}\right)$$

This equation can be implemented through the Coefficient form PDE interface in COMSOL 6.0 (COMSOL Inc., Burlington, MA, USA). However, the COMSOL solves all the governing equations in the Cartesian system (Equation (9)) for one dimension system.
(9)$$\frac{\partial c}{\partial t}=\frac{\partial }{\partial x}\left(D_{\mathrm{cart}}\frac{\partial c}{\partial x}\right)$$

We need to modify Equation (9) to implement the spherical coordinate system. Multiplying both sides of Equation (9) by *r*^2^ can give
(10)$$r^{2}\frac{\partial c}{\partial t}=\frac{\partial }{\partial r}\left(r^{2}D\frac{\partial c}{\partial r}\right)$$

Equation (10) can be implemented in the COMSOL as Equation (9) using the coefficient of ∂*c*/∂*t* as *r*^2^, *D*_cart_ = *r*^2^*D*. In this case, boundary conditions at the surface of a sphere are expressed as Equation (11). The initial conditions and the boundary conditions at the center of the sphere are defined by Equations (3) and (4), respectively.
(11)$$-n\cdot (-D\nabla c)=r^{2}h_{m}\left(c_{v,\mathrm{surf}}-c_{v,\mathrm{air}}\right)$$

### 2.3. For an Infinite Plate

For the dough sheet tested in this work, its thickness is much smaller than its length and width, then mass transfer in this infinite plate can be simplified to a one-dimension problem and described by
(12)$$\frac{\partial c}{\partial t}=\frac{\partial }{\partial x}\left(D\frac{\partial c}{\partial x}\right)$$
where *x* is the dimension from the symmetric axis to the surface of the thin film. The initial conditions, boundary conditions at the symmetric axis and the exposed surface are defined by Equations (3), (4) and (6), respectively.

### 2.4. Chemical Potential Gradient as a Driving Force

Using the principle that the mass flux is proportional to the gradient in chemical potential [21], the diffusive flux of water, *J*, is expressed as
(13)$$J=-cM_{b}\nabla \mu$$
where *M*_b_ is called mobility coefficient (m^2^ mol/J/s), and *μ* is chemical potential of water. The chemical potential is defined as the free energy per mole of water molecules. It is necessary, therefore, to be multiplied by the concentration *c* to obtain the actual free energy gradient. The chemical potential of water can be calculated using the following:(14)$$\mu =\mu_{0}+RT\ln a_{w}$$
where *μ*_0_ is the standard chemical potential. Substituting Equation (14) into Equation (13) gives Equation (15), which can be rewritten as Equation (16). By comparing Equation (16) to Fick’s first law (Equation (2)), the effective diffusion coefficient is found to be identified as Equation (17) and noted here as *D*_b_. In this sense, the diffusivity *D*_b_ is a function of mobility, water concentration and water activity.
(15)$$J=-\frac{cM_{b}RT}{a_{w}}\nabla a_{w}$$
(16)$$J=-M_{b}RTc\frac{\partial \ln(a_{w})}{\partial c}\nabla c$$
(17)$$D_{b}=M_{b}RTc\frac{\partial \ln(a_{w})}{\partial c}$$

Equation (17) can be further reformed as Equation (18), where *RTM*_b_ is often referred to as self-diffusivity, and the partial differential term is known as the thermodynamic factor [22]. Self-diffusion is the simplest diffusive phenomenon, i.e., Brownian motion of identical particles without a net flow induced by a thermodynamic gradient.
(18)$$D_{b}=M_{b}RT\frac{\partial \ln(a_{w})}{\partial \ln(c)}$$

## 3. Materials and Methods

### 3.1. Materials

Glutinous rice grain was purchased from the local supermarket. It was chosen because of its longer shape than other cereals. After cutting away its two ends, the remaining body is approximately a cylinder, making water diffusivity calculations easier. Radius of the cylinder was regarded as the average of the long and short radii of the rice grain. The average sizes of the rice cylinders were measured using digital micrometer (Syntek, Syntek Electronic Science & Technology Co. Ltd., Huzhou, China), resulting in values of 0.74 mm for the short radius, 0.93 mm for the long radius and 2.63 mm for the length.

Glutinous rice flour used for the experiments was donated by Zhengzhou Synear Food Company Ltd. (Zhengzhou, China). Its averaged radius was determined by means of image processing of its scanning electron microscope (SEM) picture, being 2.8 µm. Please refer to Reference [23] for more details.

Jinyuan^TM^ grade one flour, purchased from the local supermarket, was used for wheat dough film preparation. The dough was prepared with about 40% water content and rested for about 1 h to allow the gluten network to fully develop. A small piece of dough was kneaded by hand into a cylinder with diameter of about 4 mm. The dough cylinder was put on a smooth plastic plate and rolled back and forth along its axial direction using a stainless steel bar until a very thin film was formed. Let the film dehydrate at room temperature for about 1 h and become semi-dry. The film was sliced into squares with a side of approximately 10 mm using a sharp paper-knife. After ambient drying for another 3 h, thin dough films were obtained. Averaged thickness of the dough films was 98 mm, as measured with the digital micrometer.

In order to avoid an extended drying stage in the DVS sample chamber, all the specimens were dehydrated with P_2_O_5_ for one week at room temperature before sorption measurements.

### 3.2. DVS Measurements

A dynamic vapor sorption system (DVS Intrinsic, Surface Measurement Systems, Middlesex, UK) was used for water sorption kinetic measurements. Operating temperature was fixed at 40 °C for glutinous rice grains and dough films and 30 °C for glutinous rice flour. The humidity was controlled by mixing dry and water vapor-saturated nitrogen streams at a total flow of 200 mL/min.

#### 3.2.1. DVS of Glutinous Rice Grians

Six glutinous rice cylinders were loaded on the bottom of the sample pan and separated from each other. First, the samples were equilibrated in the DVS chamber (RH = 5%) using an equilibrium criterion of a change in mass over time (dm/dt) of no greater than 0.0003%/min for five consecutive minutes. Then, the specimens were submitted to a hydration process with RH increasing from 5 to 95% with a 10% increment, followed by desorption from RH 95% back to 5% with the same RH step size. The dm/dt criteria for hydration and desorption were the same as that for the initial drying stage. Other two tests, one from RH5% to RH85% and back to 5% with an RH step size of 20%, another from RH5% to RH 95% and back to RH5% with an RH step size of 30%, were conducted with the same dm/dt criterion as stated above. For the RH scanning DVS test, RH was linearly increased from 5% to 85% within 93 h.

#### 3.2.2. DVS of Glutinous Rice Flour

A homemade stainless steel disc of diameter 18 mm was used with the aim of carrying a thin flour layer of enough mass. The sample amount was 13–17 mg. The steel disc carrying flour was carefully put on the sample pan for DVS measurement. Please refer to Reference [24] for more details. The data from the RH0% → RH90% → RH0% sorptions with an RH step size of 10%, and the data from the RH5% → RH10% → RH15%, RH40% → RH60% adsorptions, and the RH90% → RH85% → RH80%, RH80% → RH60% desorptions were adopted in present work.

#### 3.2.3. DVS of Dough Films

A small homemade bracket made of thin iron wire was placed on the bottom of the sample pan. A piece of dough film was put on the bracket. In this way, as much surface area as possible was exposed to environmental RH during sorption measurements. The film was first dried in DVS chamber at RH 0%, then submitted to a hydration process with RH increasing from 0% to 90% with an RH increment of 10%. In another test, after the initial drying process, the dough film was submitted to a 21-step hydration process. Their RH step sizes were 2%, 2.5%, 2.9%, 3.8%, 4.1%, 5.2%, 6.4%, 7.3%, 7.5%, 7.5%, 6.5%, 6%, 5.3%, 4.2%, 3.5%, 2.7%, 2.4%, 2.1%, 1.8% and 1.7%, respectively, from low to high RH levels. The initial purpose of these random RH step sizes was to obtain an almost equal water gain in each RH step.

#### 3.2.4. Water Concentration Calculation

For the glutinous rice flour and the dough film, their masses at the end of the initial drying step during DVS tests were assumed to correspond to the mass of the dried material and were used to calculate water gain in the following sorption steps. For glutinous rice grains, separate DVS tests were conducted to equilibrate the grains at RH 5%. Water content of the equilibrated rice grains was measured after being dried in a hot air oven at 105 °C for 5 h. With initial water content known, the water gain of glutinous rice grains during DVS sorptions can be evaluated. Volumes of the rice cylinder, the rice flour particle and the dough film were calculated from their size in three dimensions. Then, water concentration evolution during each sorption was determined.

### 3.3. Model Implementation and Parameter Estimation

The diffusion model described in Section 2 was numerically solved in the COMSOL software through the Transport of Dilute Species interface for the glutinous rice grains and dough films and the Coefficient form PDE interface for the glutinous rice flour. Parameter *D* or *M*_b_ was optimized using the least squares method. The objective function (*OF*) is the sum of the squared differences between the simulated water concentration (*c*_num_) and the experimental water concentration (*c*_exp_).
(19)$$OF=\sum_{i}^{m}{\left[c_{\exp}-c_{\mathrm{num}}\right]}^{2}$$
where *m* represents the number of time steps. The *c*_num_ values were predicted according to Fick’s second law. The optimal parameter value was that corresponding to the minimum of the *OF* and was calculated using the General Optimization interface in the COMSOL 6.0 software. In this study, Monte Carlo optimization algorithm was applied using the optimality tolerance of 0.001. The initial value was specified after a previous trial and error.

Some small mathematic tricks are required to derive *M*_b_ value. First, the isotherm model (Equation (20) in Section 4.1) was plotted using 1st Opt 8.0 software (7D-Soft High Technology Inc., Beijing, China) at *a*_w_ points from 0 to 0.9 with an increment of 0.01 and gave 91 *a*_w_~*c* data pairs. Then, ln(*a*_w_)~*c* data pairs were calculated in Microsoft Excel 2010. Finally, $\partial \ln(a_{w})/\partial c$~*c* data pairs were obtained through plot derivative curve operation using Origin 8.5 (OriginLab Corporation, Wellesley Hills, MA, USA). The $\partial \ln(a_{w})/\partial c$~*c* data sets were imported into the COMSOL 6.0 software as an interpolation function. This function was called while optimizing mobility *M*_b_.

The 1st Opt 8.0 software was applied to optimize isotherm model parameters using the algorithm of universal global optimization.

## 4. Results and Analysis

### 4.1. Isotherm

An isothermal relationship is necessary for calculating the *a*_w,surf_ in Equation (6) and the $\partial \ln(a_{w})/\partial c$ in Equation (16). The isotherm model for each cereal material was established using the averaged equilibrium water concentration (*c*) corresponding to each RH from three replicates. The *c* values and their corresponding RHs were combined from all DVS tests with different RH step sizes to ensure accuracy.

The isotherm model was chosen based on its fitting performance and mathematical simplicity to facilitate the calculation of the $\partial \ln(a_{w})/\partial c$ term in Equation (16). Out of several candidates, the Park isotherm model (Equation (20)) was found to provide the best fit for our sorption data. This composite model combines the Langmuir, Henry and clustering types of water, hypothesizing that the water molecules adsorbed according to Henry’s law can make water clusters further. In this model, *c*_L_ is the Langmuir capacity constant, *k*_L_ is the Langmuir adsorption equilibrium constant, *k*_H_ is the solubility coefficient of Henry’s law, *k*_c_ is the equilibrium constant for the clustering reaction, and *n* is the mean number of water molecules per cluster. For each cereal material, the parameter values in the Park model can be found in Table 1. However, it is important to note that the equilibrated water concentration from a DVS measurement may be slightly lower than that from a saturated salt solution method, especially at low RH levels [25].
(20)$$c=\frac{c_{L}k_{L}a_{w}}{1+k_{L}a_{w}}+k_{H}a_{w}+k_{c}a_{w}^{n}$$

### 4.2. Observation from Glutinous Rice Grains

The measured water concentrations and their predicted values during some selected adsorption stages of glutinous rice grains are presented in Figure 1. For the adsorption with an RH step size of 10%, Fick’s second law provided an acceptable prediction. With the RH step size increase, the fitting performance became bad, particularly in the initial stage. Fick’s second law overestimated water concentrations in the beginning and underestimated them in the middle, especially for the adsorption with a greater RH step size. These results align with earlier studies of rice grains [4], bread crust [18] and wheat flour [19] that also used a 10% RH step size. The reaction–diffusion approach can improve modeling performance [23].

The diffusivities estimated according to Fick’s second law are shown in Figure 2. Their values ranged from 0.24 × 10^−11^ m^2^/s to 2.89 × 10^−11^ m^2^/s, depending on RH level and RH step magnitude. The water diffusivities reported in the literature for rice endosperm were summarized in Table 2. Our results are consistent with the values from sorption methods [26,27]. During desorption, water diffusivities in glutinous rice grains were lower than in adsorption at the same RH level and RH step size (see Figure 2), which is different from the observation of Prakash et al. [4]. Moreover, in this report, the maximum diffusivity during desorption is reached at 60–80% RH levels.

As shown in Figure 2, the effective diffusivity derived from Fick’s second law presents an increasing trend with RH increase for both adsorption and desorption, no matter whether RH step size was 10%, 20% or 30%. The literature reveals that the diffusivity variation pattern with RH, as observed from DVS measurements, can be classified into three categories: skewed bell-like curve (e.g., durum semolina [17] and bread crumb [6]), remaining almost constant then decrease (e.g., wheat flour and its major components [19]), monotone increase (e.g., bread crusts [18], starch and gluten films [7]). Water concentration-dependence of diffusivity for parboiled rice [11] and peas [15] from the drying method and for cracker particles from the sorption method [33] falls into the third group.

An interesting finding was that the diffusivities from the DVS tests with a greater RH step size were higher than those with a smaller RH step size; this difference weakened at lower RH levels. In theory, the diffusivity of a material should be unaffected by the testing conditions. We will discuss this further in Section 5. It can be seen from Table 2 that the effective diffusivities during soaking and drying are typically higher than those during adsorption and desorption. This may be attributed to the small RH jump during sorption measurements. Here, it has to be pointed out that the results of Steffe and Singh [26,27] were obtained with the assumption of instant equilibrium boundary conditions, and if convective interfacial mass transfer resistance is factored in, greater values will be obtained.

### 4.3. Observation from Glutinous Rice Flour

The diffusivity values of glutinous rice flour particles (Figure 3) were found to be similar to those reported for starch (~10^−16^ m^2^/s) [19]. When water adsorption is considered as a reaction–diffusion process, the diffusivity of free water in glutinous flour particles is about one order of magnitude higher than the present results [23]. Interestingly, diffusivities at low relative humidity (RH) levels were higher than those at high RH levels, in contrast to the trend observed in glutinous rice grains (Figure 2), similar to the results of the main components of wheat flour [19]. This observation can be attributed to the heterogeneity of the microstructure of rice grains, which contain abundant micro-pores. Then, other water transfer mechanisms, such as capillary flow, can be involved at high RH levels, and all are lumped into the effective diffusivity in the frame of the Fickian diffusion, resulting in a higher diffusivity. In contrast, glutinous rice flour particles are mainly composed of starch granules, which have a dense structure. As a result, water clustering at high RH levels can hinder the movement of water molecules in the dense solid matrix, leading to lower diffusivity. Interestingly, diffusivities during adsorption were observed to be greater than those during desorption at the same RH level, which can be explained by their different energy demand. Isosteric heat for water desorption was found to be lower than that for water adsorption for both rice [34] and wheat [35]. Additionally, the diffusivities from adsorption with a greater RH step size were higher than those with a smaller RH step size, and the difference was more significant for adsorption than desorption (Figure 3).

### 4.4. Observation from Wheat Flour Dough Films

Diffusivities of water in dough films are shown in Figure 4. The diffusivity from the RH step 10% measurement increased slowly and almost linearly from 1.46 × 10^−13^ m^2^/s to 1.08 × 10^−12^ m^2^/s as RH increased from 5% to 65%, then rapidly increased to a final value of 2.93 × 10^−12^ m^2^/s at RH 85%. For the 21-step DVS measurement, diffusivity remained relatively constant until RH reached 20%, then slightly and consistently increased with a further increase in RH. However, there was no significant increase observed when RH exceeded 70%. After comparing the results from these two multi-step DVS tests, one can conclude that, even for the wheat flour dough films, a greater RH step size also gave a higher diffusion coefficient. This difference was more significant in the RH range higher than 65% (Figure 4). In a related study, it was observed that water diffusivities of the starch film (104 µm thickness) and gluten film (198 µm thickness) at 25 °C increased from 0.1 × 10^−13^ m^2^/s to 1.2 × 10^−13^ m^2^/s and from 0.1 × 10^−12^ m^2^/s to 2 × 10^−13^ m^2^/s, respectively, as RH increased from 10% to 80% [7].

### 4.5. Observation When Chemical Potential Gradient Was Considered as a Driving Force

The estimated values for *M*_b_ are presented in Figure 5. Different from the effective diffusivity shown in Figure 2, all the water mobility curves were concave upward throughout the RH range, suggesting a more rapid increase than the diffusivity, particularly at high RH levels. The water mobility was found to be dependent on RH step magnitude, with the discrepancy being more significant at middle to high RH levels.

By multiplying the *M*_b_ values shown in Figure 5 by *RT* (= 2604 J/mol at 40 °C), we can calculate the self-diffusivities in glutinous rice grains using Equation (18). The resulting values ranged from 1.34 × 10^−12^ m^2^/s to 4.31 × 10^−11^ m^2^/s. Water self-diffusivity of polished rice after soaking at 25.0 °C for 30 min is 1.4 × 10^−14^ m^2^/s [36]. The same coefficient of glutinous rice flour and rough rice is in the order of magnitude of 10^−11^ m^2^/s [37] and 10^−12^ m^2^/s [38], respectively. Those two studies derived their diffusivities from isotherm data, where the spreading pressure gradient was considered the driving force for diffusion and water flux was expressed as [38] J=−ηρm∇π, where *π* is the spreading pressure, *η* is a resistance coefficient, *ρ* is the solid density and *m* is the water content (db), then ρm=c. Comparing with Equation (13) suggests that the two *J* expressions in terms of *π* or *μ* are actually equivalent when *η* is considered as *M_b_*. Since the *RT* values are equal for each DVS test, it is obvious that the self-diffusivities are also unidentical when different RH step sizes are applied.

With the $\partial \ln(a_{w})/\partial c$~*c* data pairs obtained as described in Section 3.3, we can calculate the variation of *D*_b_ with *c* using Equation (17), and present in Figure 6. It is evident that the variation of the averaged diffusivity in each adsorption stage with water concentration follows the trend of diffusivity from the adsorptions with an RH step of 10% (refer to Figure 2). As we increase the water concentration, diffusivity slowly then rapidly increases. Now, let us examine each adsorption step closely. During the water adsorption stage of RH5% → RH15%, the diffusivity slightly increased as the water concentration increased. However, for the last seven adsorption steps, diffusivity always decreased with the water concentration increase, which is the opposite trend of the averaged diffusivity. This is another discrepancy in diffusivity when the potential chemical gradient acts as a diffusion driving force. The values of *M*_b_ and *RT* are constant for a specific adsorption stage, then the variation trend of *D*_b_ only depends on $\partial \ln(a_{w})/\partial \ln(c)$ (see Equation (18)), which is dominated by the isotherm relation (Equation (20)). The inset of Figure 6 shows the calculated values of $\partial \ln(a_{w})/\partial \ln(c)$, which increased as the water concentration rose until to ~50 kg/m^3^, then decreased as the water concentration increased further. This can elucidate the opposite trend of the *D*_b_ in the last several adsorption steps to the first step.

### 4.6. Observation from RH Scanning DVS Measurement

The RH scanning DVS test, which can be used as an alternative method for determining glass transition [39], can also provide insight into the water diffusion rate during sorption. In contrast to traditional DVS methods, this procedure offers numerous benefits, such as almost eliminating the impact of the external mass transfer coefficient on the diffusivity value [40]. Figure 7 illustrates the water concentration evolution during the RH scanning measurement of glutinous rice grains. The adsorption curve can be divided into four regimes. Initially, the adsorption rate was very low, followed by a rapid increase. When approximately half of the water gain was achieved, the adsorption gradually slowed down. At the final stage, the adsorption sped up again. Reference [39] attributes the first stage to surface adsorption, after which bulk adsorption involves. We believe that this only holds true for fine particles. For large particles such as rice grains, the situation is more complicated. Upon carefully examining the diffusivity curve of the adsorption with an RH step size of 10% (refer to Figure 2), it becomes apparent that the curve can also be split into four regions.

During the RH scanning sorption measurement, the rate of RH increase is extremely slow at 0.86% RH per hour. This indicates that the difference between the RH levels close to the sample surface and in the DVS sample chamber will likely be much less than 10% [40]. Based on our previous DVS test results, where we varied the jumping RH levels, the diffusivities obtained from the RH scanning measurement are expected to be smaller than those of the adsorption with a 10% RH step size. It is challenging to extract diffusivities from the RH scanning sorption data by itself. As an alternative, we imported the diffusivity–concentration data pairs from the DVS test with a 10% RH step size (displayed in Figure 2) into the COMSOL as an interpolate function. This allowed us to predict water concentration during the RH scanning adsorption, which is shown in Figure 7. Notably, the predicted water concentrations were consistently higher than the measured values, especially during the initial half of the adsorption duration. So, just as anticipated, the water diffusivity during the RH scanning measurement was smaller than that from the adsorption of RH step size 10% at equivalent RH levels. This indirect support confirms our earlier observation of diffusivity discrepancy from different RH step sizes.

## 5. Discussions

### 5.1. Measurement Condition-Dependence and beyond of Water Diffusivity

Various mechanisms, such as capillary flow, hydraulic flow and evaporation–condensation, are involved in water transfer in biopolymer-based materials [1,41]. However, the Fickian approach models all the possible mass transport mechanisms using a single Fick-type flux. Our results suggested that Fick’s second law, although widely used in cereal research (see Table 2) and able to give an acceptable fit to water sorption kinetics, is only a phenomenological description. Caution is recommended when extrapolating the effective diffusivity obtained under one sorption condition to another. For example, a difference in RH step size can violate the extrapolation.

In theory, diffusivity should be dominated only by a material’s nature, but our research showed that water diffusivity depends on measurement conditions. Previous studies have found that effective water diffusivity from Fick’s second law depends on particle size [17], film thickness [7], initial water concentration, RH of drying air and thickness of the sample [42]. Water diffusivity varied with initial water concentration and RH of drying air because these factors alter the material properties such as microstructure [41] and size [31] simultaneously as they cause water transfer. Thickness–dependence of water diffusivity is well documented in the literature for synthetic polymer [43] and biopolymer [44] materials, which can be explained in terms of polymer confinement [41] or a two-region model [45]. Therefore, the thickness–dependence of water diffusivity is understandable, as all these proposed explanations point to property modification accompanied by thickness variation.

Our research showed that different RH step sizes in the same RH range could result in different diffusivity, which can not be simply attributed to the property modification during sorption processes. Even if the cereal materials were modified during sorption measurements, the histories that the samples experienced during two test procedures, one with two continuous adsorptions with a small RH step (e.g., 25% → 35% → 45%), the other with a large RH step (e.g., 25% → 45%), will be very similar.

### 5.2. Future Research Suggestions

We tried our best to find similar reports in the literature to check our observations. However, there is limited data available on this subject, with no systemic research published yet. This could be because the DVS system has been widely used only in recent years. We did find a related study on the Nafion^TM^ membrane, which reported effective water diffusivities during vapor adsorptions under different initial–final RH combinations [46]. Interestingly, parts of their findings supported our observations. For instance, they reported that the diffusivities from RH20% → RH40% and RH40% → RH80% adsorptions at 60 °C were 1.42 cm^2^/s and 2.56 cm^2^/s, respectively. Similarly, the diffusivity from RH20% → RH80% adsorption was measured as 2.92 cm^2^/s [46]. Of course, a contrasting trend was also observed in this study. Moving forward, we hope to see more studies on this topic and extend our observations to other types of materials.

We recognize that diffusivity is often evaluated based on the evolution of space-averaged water concentration with time, making it an apparent value. Other techniques, such as time domain–nuclear magnetic resonance [47] and in situ time-resolved Fourier transform infrared spectroscopy [48], calculate the water diffusion coefficient from water distribution in space, which can provide true diffusivity [49]. We believe that examining the discrepancy we observed between apparent and true diffusivities is the first step toward future research. This analysis will help us better understand the role played by RH step size in water sorption, thereby informing the development of a phenomenological model that can explain this discrepancy and suggest practical applications.

It is challenging to develop such a diffusion model to address this confusing discrepancy. The difficulty stems from understanding how to incorporate the effects of RH step size into the conservation equation in addition to boundary conditions. In the traditional Fickian framework, the boundary condition approach is the only way to consider the contribution of external resistance to mass transfer. We resorted to the concentration gradient-dependent diffusivity model proposed by Kruckels [50] and the Fokker–Planck law of diffusion, which includes an additional mass flux equal to c⋅∂D(x,t)/∂x in its conservation equation [51], with the aim of getting a consistent diffusivity. However, all our efforts failed. Understanding the routine through which RH step size affects water sorption is considered to be a prerequisite for discerning the right model or proposing a new one.

The effective diffusivity of glutinous rice grains is significantly higher than that of glutinous rice flour (refer to Figure 2 and Figure 3). This result suggests that structural heterogeneity in tissue or cell level may play an important role in determining water diffusion. Previous research has shown that the diffusion and distribution of water within cereal grains are regulated by cell walls in the aleurone layer [52], and these plant tissue microstructures can be modeled numerically [53]. Recent developments in multiphase [54] and multiscale [55] modeling of water transfer have provided additional insight into this phenomenon. It is hoped that these new approaches will shed light on the confusing diffusivity discrepancy.

Our results showed that even for glutinous rice flour, the diffusivity discrepancy was still observed. So, other factors than the microstructure may contribute, at least partially, to the discrepancy. For example, dielectric permittivity [56] and stress [57] relaxations can indirectly impact water diffusion, implying that dielectric and stress properties under different RH step sorption procedures warrant further exploration. Notably, diffusivity during desorption is lower than that during adsorption. Regardless of the sorption–history effects, additional refinement to the traditional modeling approach, which consists of diffusion in the body plus convective mass transfer at the boundary, is necessary. All in all, water adsorption is a complex phenomenon that involves at least four successive processes, including water vapor convection in the bulk gas phase, diffusion in the solid–gas interfacial stagnant layer, adsorption on the particle surface and molecular diffusion into the inner region of the solid matrix [58].

## 6. Conclusions

After analyzing the isothermal sorption kinetic data of glutinous rice grains, glutinous rice flour and wheat flour dough films according to Fick’s second law, we have discovered that a greater RH step size results in a higher diffusion coefficient even in the same RH level, regardless of whether water concentration gradient or potential chemical gradient is considered as the driving force for diffusion. This finding suggests caution when using the diffusivity values obtained from DVS measurements in engineering designs.

The observed discrepancy can not be simply attributed to the condition-dependence of water diffusivity alone. It is important to note that in most cases, Fick’s second law only serves as a phenomenological description of a kinetic sorption process. To confront this issue, a diffusion model, whether phenomenological or mechanistic, must be developed to address or eliminate the discrepancy. This development is crucial to both practical and theoretical applications.

## Figures and Tables

**Figure 1 foods-12-01470-f001:**
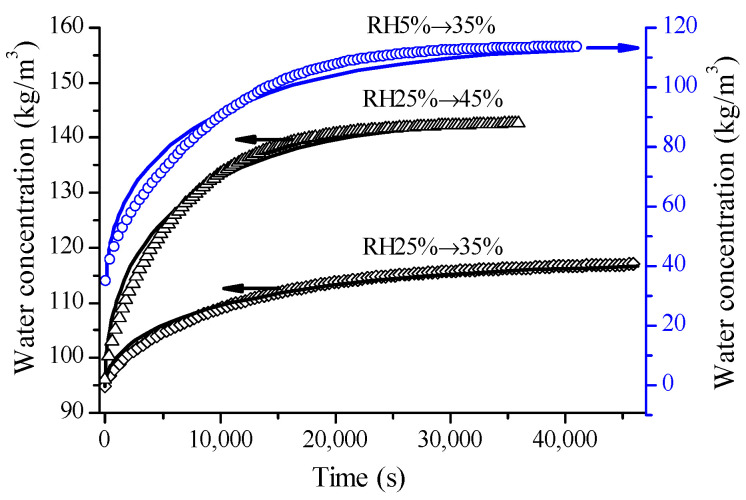
Water concentration evolutions of glutinous rice grains during the adsorptions of RH25% → 35%, RH 25% → 45% and RH5% → 35% (symbols) and their predictions according to Fick’s second law with their estimated diffusivities (lines).

**Figure 2 foods-12-01470-f002:**
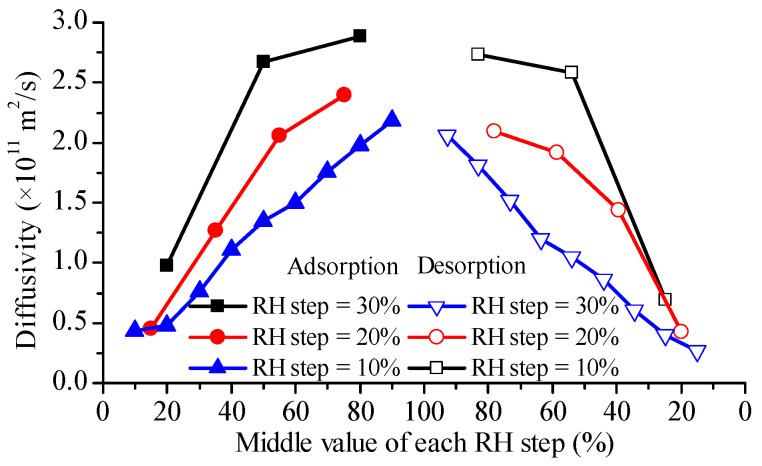
Water diffusivities in glutinous rice grains during adsorptions and desorptions with different RH step sizes obtained according to Fick’s second law.

**Figure 3 foods-12-01470-f003:**
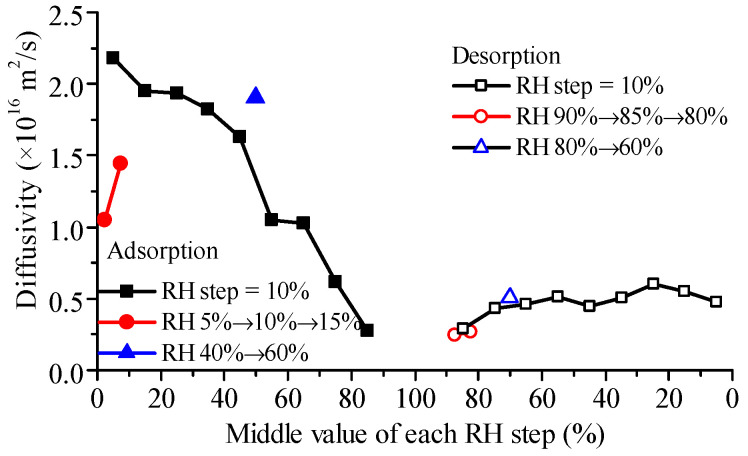
Water diffusivities in glutinous rice flour particles during adsorption and desorption with different RH step sizes obtained according to Fick’s second law.

**Figure 4 foods-12-01470-f004:**
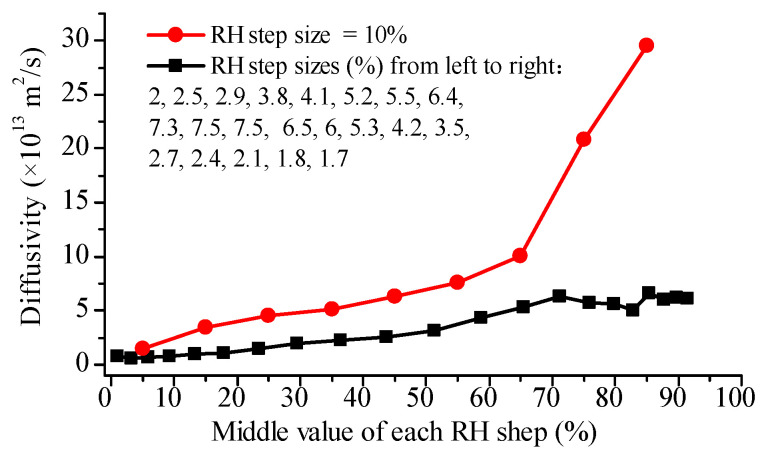
Water diffusivities in dough films during adsorption with different RH step sizes obtained according to Fick’s second law.

**Figure 5 foods-12-01470-f005:**
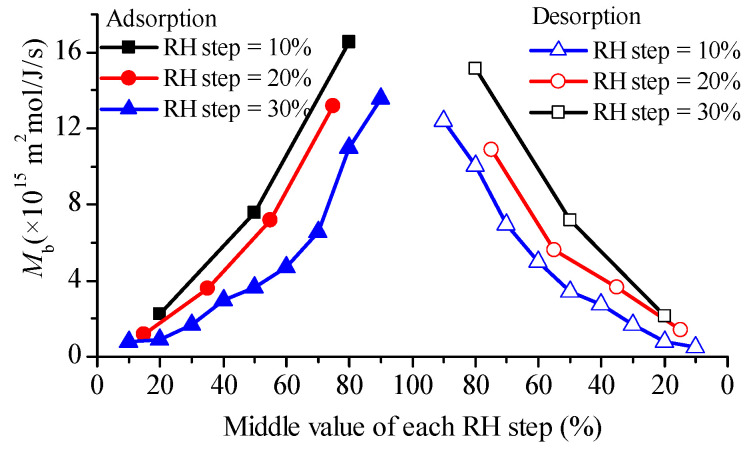
Water mobilities in glutinous rice grains during adsorption and desorption with different RH step sizes obtained according to Fick’s second law when potential chemical gradient was considered as a driving force.

**Figure 6 foods-12-01470-f006:**
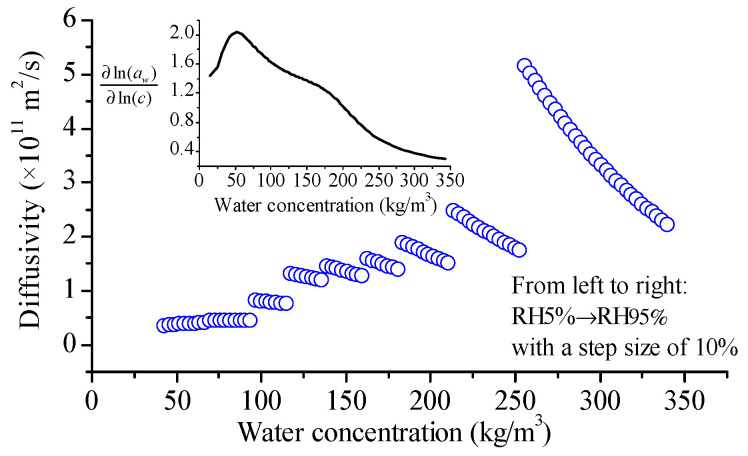
Variation of water diffusivity in glutinous rice grains with water concentration during the adsorption with an RH step size of 10%. Here, the diffusivities were obtained according to Fick’s second law with potential chemical gradient as a driving force for water diffusion.

**Figure 7 foods-12-01470-f007:**
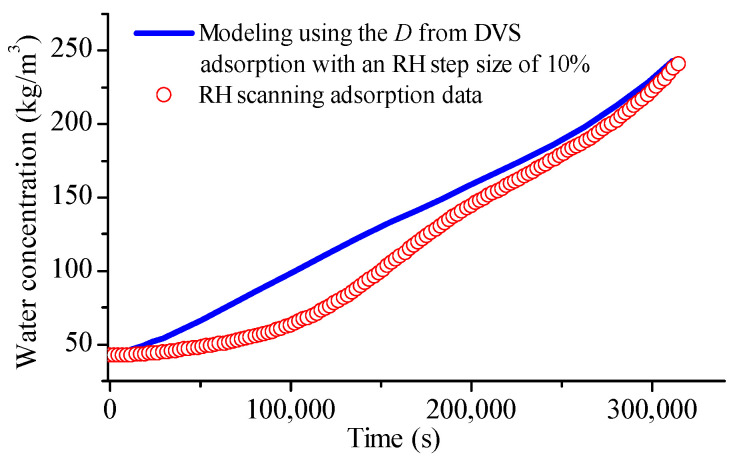
Evolution of water concentration of glutinous rice grains with time during the RH scanning DVS measurement and its prediction using the diffusivities from the DVS test with an RH step size of 10% according to Fick’s second law.

**Table 1 foods-12-01470-t001:** Parameter values in the Park isotherm model for the three cereal materials.

	Adsorption						Desorption					
	*c* _L_	*k* _L_	*k* _H_	*k* _c_	*n*	R^2^	*c* _L_	*k* _L_	*k* _H_	*k* _c_	*n*	R^2^
Glutinous rice grains	46.12	39.94	218.44	172.21	10.15	0.9930	63.45	21.17	220.64	117.67	8.62	0.9968
Glutinous rice flour	72.58	12.91	188.17	164.20	6.85	0.9998	93.99	11.86	193.47	96.82	5.48	0.9988
Wheat dough films	51.84	11.01	197.96	295.39	11.28							

**Table 2 foods-12-01470-t002:** Reported effective diffusivity values of water in rice endosperm.

Experimental Method	Diffusion Model	Grain Type	Diffusivity (×10^11^ m^2^/s)	Notes	Ref.
Drying	Fick’s 2nd law + DBC	Short grain	7.24	*MC*_i_ = 29.8% db; 40 °C RH of drying air = 68.6%	[26]
Drying	Fick’s 2nd law + DBC	Short grain	6.27	*MC*_i_ = 34% db; 40 °C RH of drying air = 63.4%	[27]
Drying	Fick’s 2nd law + CBC	Long grain	14.10	*MC*_i_ = 19% db; 40 °C	[28]
Adsorption	Fick’s 2nd law + DBC	Long grain	~2.2	*MC*_i_ = ~13.2% db RH = 84–90%; 40 °C	[29]
Adsorption	Fick’s 2nd law + CBC	Medium grain	1.11, 0.19, 0.20, 0.80, 0.49	RH 0% → 20% → 40% → 60% → 80% → 98%; 25 °C	[4]
Desorption	Fick’s 2nd law + DBC	Medium grain	0.4, 0.80, 1.31, 1.51, 1.30	RH 98% → 80% → 60% → 40% → 20% → 0%; 25 °C	[4]
Soaking	Fick’s 2nd law + CBC	Long grain	6.67	*MC*_i_ = 14.5% db; 40 °C	[30]
Soaking	Fick’s 2nd law + DBC	Long grain	16.20	*MC*_i_ = 17–22% wb 40 °C	[31]
Soaking	Fick’s 2nd law + DBC	Long grain	8.62	*MC*_i_ = 16.5–22.5% wb 40 °C	[32]

Notes: DBC: Dirichlet boundary conditions; CBC: convective boundary conditions; *MC*_i_: initial moisture content; db: dry base; wb: wet base.

## Data Availability

Data is contained within the article.

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
