# Peer review of "Discrepancy of Effective Water Diffusivities Determined from Dynamic Vapor Sorption Measurements with Different Relative Humidity Step Sizes: Observations from Cereal Materials"

_foods, 2023, doi:10.3390/foods12071470_

Round 1

Reviewer 1 Report

Overall, this work is valuable, and the results and discussion are rich.

However, the English, in most part of the text, cannot be accepted in a professional context. Apart from many grammatical errors, the authors wrote words that do not even exist (phenomenical, tailared, inhomogenesity, permittivity).

Also, the authors do not follow the sections they used. In my opinion, the descriptions of the mathematical models should be the material and methods section or in another section dedicated to those but not in the results section.

Sometimes, the speech is too informal: “as we all know”, “we will give a dep discussion on this topic at the end of this paper”, “our limited suggestions”.

Some words exist but are used incorrectly: “imagine” instead of “image”, “try and error” instead of “trial an error”, alterative, “please bearing in mind”

The authors refer to “optimized diffusivity” while they mean “estimated diffusivity”. Also “parameter optimization” should be substituted by “model fitting” or “estimation of mass transfer parameters”.

The authors should try more to provide explanations of why “a greater RH step size will result in a higher diffusivity coefficient even in same RH level”. The D are average D for a RH interval. This means that they are obtained from the integral of D along the RH interval. The authors should calculate the derivative of the average D vs time to obtain the real D.

Overall, I agree with global conclusion that Fickian diffusion model with a convective boundary layer is insufficient to explain this and other water transfers, though it is by far the most used model.

DETAILED OBSERVATIONS:

 Section 2.2:

Why the authors used “40 °C for glutinous rice grains and dough films, and 30 °C for glutinous rice flour?

Identify the ranges in the same way, like “5 to 95%”.

Why 5 to 85% and not 95%?

Third paragraph: Why these ranges?

L124:

0 to 90% or 95%?

L154:

Do you want to mean replicates instead of materials?

L158:

Correct to Chung-Pfost.

Table1:

Decrease number size or enlarge columns width to have a number per line.

L209:

Identify the stages.

Figure 1:

Improve the legend identifying each of the curves.

Line 218:

Correct to 10ª(-11).

L221:

Figure 2 seems to show the opposite «of what the authors state here.

L227:

Fick’s second law.

L272:

Explain this opposite trend.

Fig 4:

Why was the desorption curve not presented?

Fig.6:

Only adsorption? Explain the why it decreased with increasing concentration. Identify the RH intervals.

Section 4.2 must be less formal. Sounds like a diary.

Author Response

Dear reviewer

    Thank you very much for your very helpful comments and suggestions.

    We had revised our manuscript very carefully. Two colleagues with abroad research experience also helped us improve English writing.

    The comments is replied to one by one in the following parts

Best regards

Authors

Reviewer 2 Report

The article submitted by Zhao et al. shows a method to correlate water diffusivities using DVS from cereals. The article discusses a methodology potentially impacting the agro-food sector. However, some amendments and clarifications are strongly suggested to lift the work's impact and increase the readers' attention. The comments are disclosed point-by-point below:

1) The title is long and does not fully reflect the scope of the work. What is the discrepancy? Are the authors actually studying cereal materials? 

2) How extrapolated are the results presented to other types of cereals besides the one tested in the work? Especially when the authors discuss the results being correlated to cell structure inhomogeneities (Lines 454-455).

3) How are the experimental designs correlate to real field conditions?

4) From the abstract and introduction, it is unclear what the authors are measuring/improving with the presented method. The novelty of the work should be clearer in the re-submitted version.

5) Introduction. Line 31. The problem is not clear. How much can certain ranges of RH affect agro-food processes? How can we measure the impact of the work on the current industrial processes if the hypothesis of the work is solved/implemented?

6) Introduction. Line 42. What about the dimensions of the cereal grain? It has been shown that dimensions (thickness) can considerably impact the sorption/desorption of bio-based products. The same reasoning goes for the arguments stated in Lines 289-292. Suggested literature: https://doi.org/10.1002/mame.202200630

7) Line 119. Check the spelling of "desorption." 

8) Materials. The re-submitted manuscript should include pictures/schemes of the experimental process. 

9) Figure 1. What are the differences between the two y-axes? 

10) Figure 6. What is the difference between the different curves? A legend should be included.

11) Line 472. What "four successive processes" are the authors referring to?

Author Response

Dear reviewer

   Thank you very much for your very helpful comments and suggestions.

    We had revised our manuscript very carefully. Two colleagues with abroad research experience also helped us improve English writing.

    The comments is replied to one by one in the following parts.

Best regards!

Authors

Round 2

Reviewer 1 Report

The manuscript was improved in a rush, but still cannot be published due to the English. Still, many errors persist: e.g., alterative, grians, ric, eapecially, size. Grammar and lack of care are the problems.

Also, a better framing of the different studies that were placed here together to achieve a coherent whole is suggested. It may need additional experiments or take from the manuscript part of them.

Detailed comments:

L172

Explain here why these temperatures, not just describing "this is what we did".

Section 4.1

How were the isotherms obtained? Which RH steps were used to obtained them? Were there any differences due to the RH step?

Table 2

Still not homogenous and with numbers broken in two lines.

Figure 1

Why a curve that starts at RH5% has a starting water content than one starting at RH25%?

Prdications?

This curve is of which product? We can see in the text, but it should be in the legend.

Figure 3

Why the isolated values? Didn’t other values correspond to expected results and were deleted from the plot? You should do more conditions.

L323

Were diffusivities in desorption lower because more energy is needed? Discuss.

L363

Not clear what the authors mean.

L372

Explain why “opposite to the overall trend of the averaged diffusivity”. 

L377

Experimentally, what is the difference to the other previous measurements? 

Author Response

Dear reviewer,

      Thank you again for your very careful and insightful suggestions. We had revised our manuscript, and answered your comments in the following Word file. It is hoped that this is a satisfactoy answer.

      Best regards!

Authors

Reviewer 2 Report

The authors have responded to the issues raised by the reviewer. Check minor grammar spellings in lines 349 and 354. The article is thereby ready to be accepted. 

Author Response

Dear reviewer,

      Thank you very much for your helpful suggestions. Our responses to the comments is attached in the following file.

      Best regaeds!

Authors
